# Effect of Structure of Carbonized Polymer Dot on the Mechanical and Electrical Properties of Copper Matrix Composites

**Jiahui Xu [1,2], Wenmin Zhao [1,*], Shaoyu Li [1], Rui Bao [2], Jianhong Yi [2,*] and Zhaojie Li [2]**

[1] Faculty of Materials Metallurgy and Chemistry, Jiangxi University of Science and Technology, Ganzhou 341000, China
[2] School of Materials Science and Engineering, Kunming University of Science and Technology, Kunming 650093, China
* Correspondence: 9120220038@jxust.edu.cn (W.Z.); yijianhong@kmust.edu.cn (J.Y.)

**Abstract:** Carbonized polymer dots (*CPD*s) have been paid a lot of attention by researchers because of their excellent properties due to their unique structure. However, few studies have researched the impact of the *CPD* structure on composite applications. Herein, *CPD* and heat-treated *CPD* (*h-CPD*) are used to fabricate a reinforced Cu matrix composite. There was a semi-coherent interface between *h-CPD* and Cu. However, functional groups and polymer chains of *CPD*s decomposed during heat treatment, weakening the mechanical and electrical properties of the composites. These findings show that *CPD* structural integrity is vital to composites.

**Keywords:** *CPD*; *h-CPD*; Cu matrix composites; structural integrity





## 1. Introduction

When it comes to determining the performance of carbon nano-material (CNM)/Cu composites, the most important aspects are good dispersion and strong reinforcement–matrix interfacial cohesion, both of which are the foci of study in the area of Cu-based composites [1]. Many effective methods have been developed for fabrication of CNM/Cu composites with good dispersion and tight interface bonding.

For example, by adjusting the acid-treatment time of Carbon nanotubes (CNTs) to produce carboxyl and other ionogenic groups, Yang et al. found that uniformly distributed CNT was obtained by extending the acid-treatment time [2]. Chu reported a new design for producing defects on the graphene (GR) basal-plane to optimize the interface and mechanical properties of GR/CuCr composites by the formation of $Cr_7C_3$ at the basal-plane/matrix interface [3]. Carbonized polymer dots (*CPD*s), which have structural characteristics of GR and CNT, as well as rich surface functional groups and water solvent dispersion, have been employed to reinforce a Cu-based matrix as a nano-carbon reinforcement [4]. These *CPD*s for Cu-based composite application, however, have polymer chains and oxygen-containing functional groups that are introduced during the preparation or when raw materials, which makes them more reactive than perfect CNT and GR, leading to the interfacial structure and bonding state between *CPD* and Cu matrix being affected.

Here, in this paper, effort was made to reveal how the structural characteristics of *CPD* affect the mechanical and electrical properties of Cu-based composites. We prepared heat treated *CPD* (*h-CPD*)/Cu composites by combining ball milling with spark plasma sintering (SPS). Moreover, the mechanical properties, electrical conductivity, microstructure and strengthening, and the conductivity mechanism of the *h-CPD*/Cu composites were carefully compared, in detail, with those of *CPD*/Cu composite prepared in the same way. It was found that the functional groups and polymer chains of *CPD* were beneficial to the Cu matrix, imparting outstanding mechanical and electrical properties.

## 2. Experiment

*CPD* was synthesized by hydrothermal means, the details of which are in our previous work [5]. In this work, the *h-CPD* was obtained by heat treatment of *CPD* at 300, 500, 700 and 900 °C, respectively. The products were labeled as *h1-CPD*, *h2-CPD*, *h3-CPD* and *h4-CPD*. It is worth noting that *h-CPD* cooled with the furnace immediately after heating to a specific temperature, and the whole process was carried out in an atmosphere of Ar. Then, the *h-CPD*s were dissolved in anhydrous ethanol and ball milled with pure Cu powders (ball milled parameters: 100 rpm 8 h + 200 rpm 5 h, and the mass fraction of the *h-CPD* was 0.2%). Finally, the composite powders were reduced (300 °C in a $N_2$-10% $H_2$ mixed atmosphere, for 6 h) and sintered by SPS in a vacuum environment at 550 °C for 5 min to obtain *h-CPD*/Cu composites. The *CPD*/Cu and Cu matrix was prepared under the same experimental conditions.

The *h-CPD* microstructures were studied by transmission electron microscopy (TEM, Tecnai G2-TF20, S-Twin, FEI company, Hillsboro, OR, USA), and X-ray photoelectron spectroscopy (XPS, K-Alpha+, Thermo Fisher Scientific, Waltham, MA, USA). The interface microstructure of the composites was analyzed by TEM. A tensile test was conducted with AG-IS 10 KN equipment (Shimadzu Corporation, Kyoto, Japan) with a crosshead speed of 0.5 mm/min at room temperature. Electrical conductivity of the composites was tested by means of a digital conductivity meter (Sigma 2008B, Xiamen Tianyan Instrument Co., Ltd., Xiamen, China). The fracture surface of *h-CPD*/Cu composites was analyzed via scanning electron microscopy (SEM, Nova Namo-450, FEI company, Hillsboro, OR, USA).

## 3. Results and Discussion

### 3.1. Microstructure

Figure 1a,b reveals the microstructure of *CPD* synthesized by hydrothermal means, which had a near-spherical distribution, with a lattice spacing of around 0.21 nm and a mean diameter of approximately 3.0 nm. Contrary to *CPD*, *h1-CPD* exhibited a thick lamellar morphology and nanoparticles, suggesting that after heat treatment at 300 °C, some *CPD*s were still intact and combined to form nanoparticles (Figure 1c). The diameter of the nanoparticles observed increased as shown in Figure 1d. This was due to the fact that the *CPD*s obtained hydrothermally were not in a single, distributed form but instead preferred to attach to one another. In addition, with further magnification of *h1-CPD*, it was discovered that certain lamellar structures exhibited an amorphous structure, while others showed distinct lattice fringes (~0.25 nm) (seen Figure 1d) and had similar morphology and structure to that of *CPD*. The increase of amorphous area in *h2-CPD* lamellar structure suggested that following heat treatment at 500 °C, most *CPD* structures were damaged (see Figure 1e,f). Interestingly, there was homogeneous distribution of nanoparticles in the *h3-CPD* thin lamellar structures. However, when the heat treatment temperature was raised to 900 °C, there were only a few nanoparticles along the *h4-CPD* flakes, and their size and quantity decreased. This was attributed to the fact that, after heating at 300 °C, cross-linking occurred between *CPD* molecules to avoid collapse, and carbonization proceeded as the temperature rose to 500, 700 °C, resulting in the formation of a self-supporting nitrogen-doped carbon lamellar structure. As the temperature continued to increase to 900 °C, the three-dimensional nitrogen-doped carbon lamellar structure was further carbonized and destroyed, giving birth to a lot of amorphous carbon. The lamellar thickness, smoothness, and diameter of *h-CPD* fluctuated dramatically when heat-treated at various temperatures. This was primarily due to variations in the stability of polymer chains and functional groups on the *CPD* surfaces.

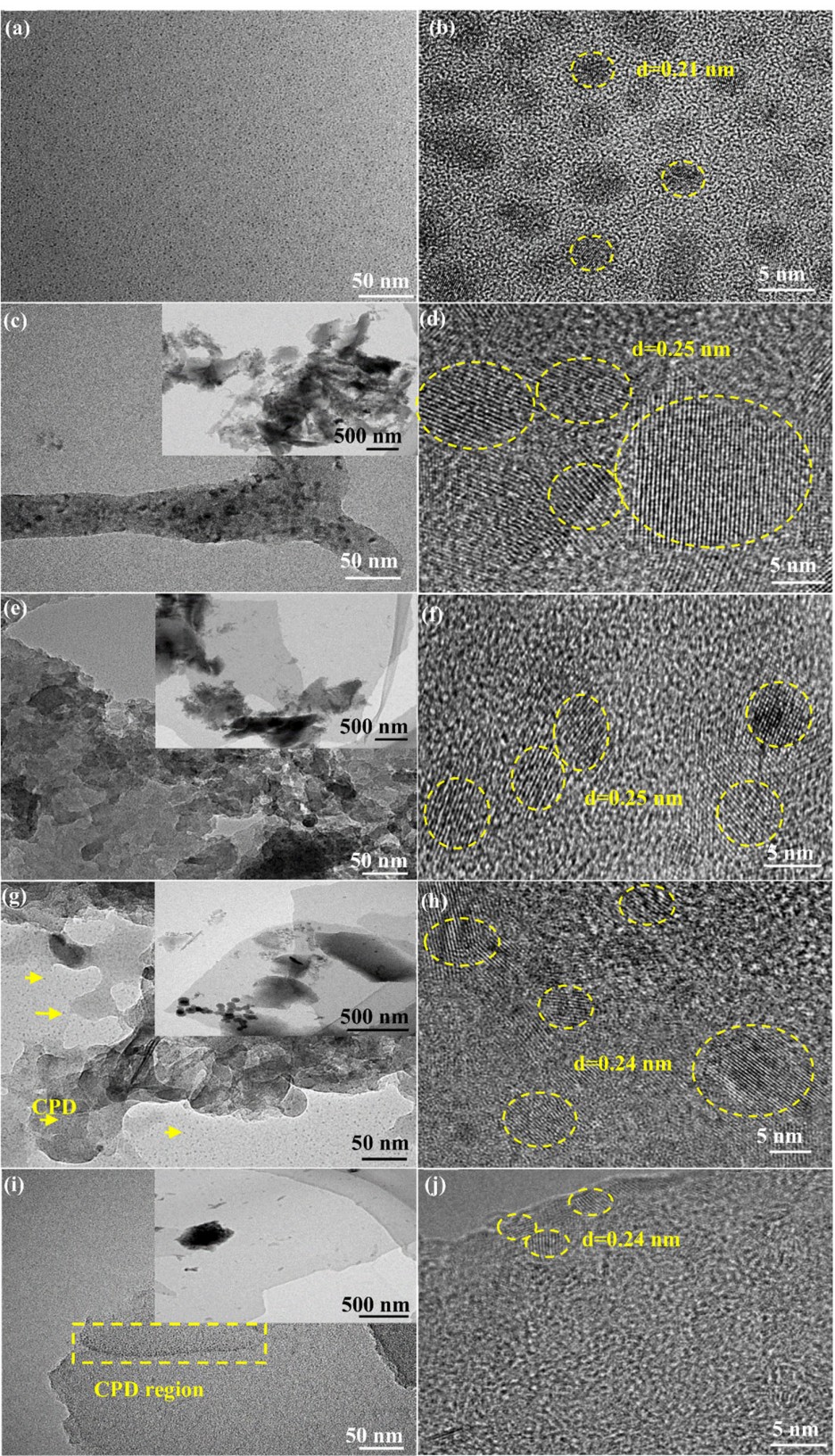

**Figure 1.** The TEM images of reinforcement: (**a**,**b**) *CPD*, (**c**,**d**) *h1-CPD*, (**e**,**f**) *h2-CPD*, (**g**,**h**) *h3-CPD*, (**i**,**j**) *h4-CPD*.

XPS was utilized to better investigate the chemical structural changes of *CPD* following heat treatment (see Figure 2a–c). All samples were carbon-rich and included carbon, oxygen

and nitrogen. The C, O and N contents of samples at various heat treatment temperatures are shown in Figure 2a. It was discovered that with the increment of heat treatment temperature, the C wt.% in *CPD* improved from 61.11% to 88.70%, O wt.% increased from 7.28% to 10.50% and then dropped to 5.22%, and N wt.% reduced from 31.25% to 6.08%. These changes implied that the hydrothermally synthesized *CPD* was carbonized again following heat treatment, and the surface polymer chains and functional groups were degraded at high temperature, leading to a drop in O wt.% and N wt.%. According to Figure 2b, as the heat treatment temperature improved in C1s, the contents of C=C, C-C, C-N/C-O, C=O/C=N, and O-C=O [5] all changed. Importantly, C=C increased significantly at first and then decreased. The C=C bond compositions of *h3-CPD* and *h4-CPD* were, however, essentially the same. This showed that when the heat treatment temperature rose to 700 °C, the polymer chain and functional groups on the *CPD* surface decomposed and the remaining C=C bonds were mostly from the C-core of *h-CPD*, and the C-core of *h-CPD* had thermal stability. Furthermore, the pattern of change in C-C bond composition was opposite to that of the C=C bond. This suggested that at high temperatures, part of the structure in *h-CPD* was destroyed and the sp$^3$ C component was generated, which was consistent with the TEM morphology of *h-CPD*. In N1s (see Figure 2c), it could be clearly observed that N-containing species in *CPD* were converted from pyrrolic-N to pyridine-N and graphite-N after heat treatment, indicating N-doping with *CPD* [6].

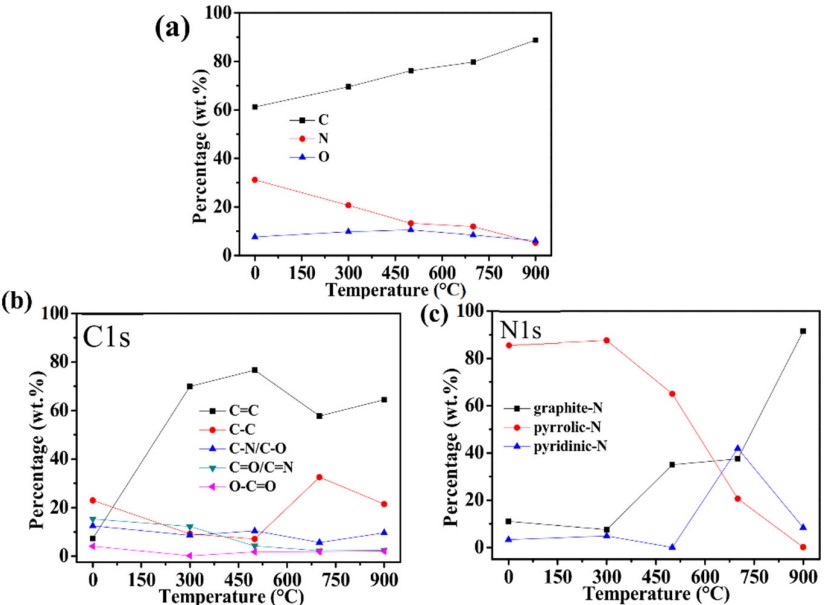

**Figure 2.** (**a**) XPS full spectrum, high-resolution of (**b**) C1s, (**c**) N1s of *h-CPD*.

Figure 3 gives the microstructure of the *h3-CPD*/Cu composite, which was different from *CPD*/Cu composites [4]. Observed under low magnification TEM, the *h3-CPD*s were discovered in the Cu grains (see yellow arrows in Figure 3a). The blue wire frame in Figure 3a is magnified in Figure 3b, and a large amount of amorphous carbon (AC) network existed in the non-matrix area, and a small amount of *CPD* with full lattice fringe still persisted in the AC network (yellow ellipse in Figure 3b), which indicated that the structure of *CPD* was destroyed after heat treatment, ball milling and SPS sintering. In addition, as shown by the white dotted line in Figure 3b, there was a clear separation between *h3-CPD* and Cu. The black dotted line portion in Figure 3b was subjected to Fourier transform (FFT) and inverse Fourier transform (IFFT) processing in order to further evaluate the interface. The findings are shown in Figure 3c,d. First of all, there was a lot of AC. Second, three groups of diffraction dots also showed up. It was found that the lattice spacings were typically ~0.21 nm, 0.24 nm and 0.24 nm, for the d-spacing (111) in the crystal plane of the Cu matrix (~0.209 nm), and the two sets of diffraction dots with a *d*-spacing of 0.24 nm

corresponded to *d*-spacing (100) in the crystal plane of *h3-CPD*. To clarify the interface structure, the misfit parameter, *ε*, was described by the following equation [7]:

$$\varepsilon = \frac{d_{h3-CPD} - d_{Cu}}{d_{Cu}} \tag{1}$$

where $d_{h3-CPD}$ and $d_{Cu}$ are the interplanar spacings of the Cu (111) and *h3-CPD* (100) planes. The calculated parameter was 0.14, which indicated that the interface between the Cu matrix and *h3-CPD* was semi-coherent. The semi-coherent interface generally exhibited a relatively low interfacial energy, which was beneficial to the formation of a strong interfacial bonding between *h3-CPD* and the Cu matrix [8].

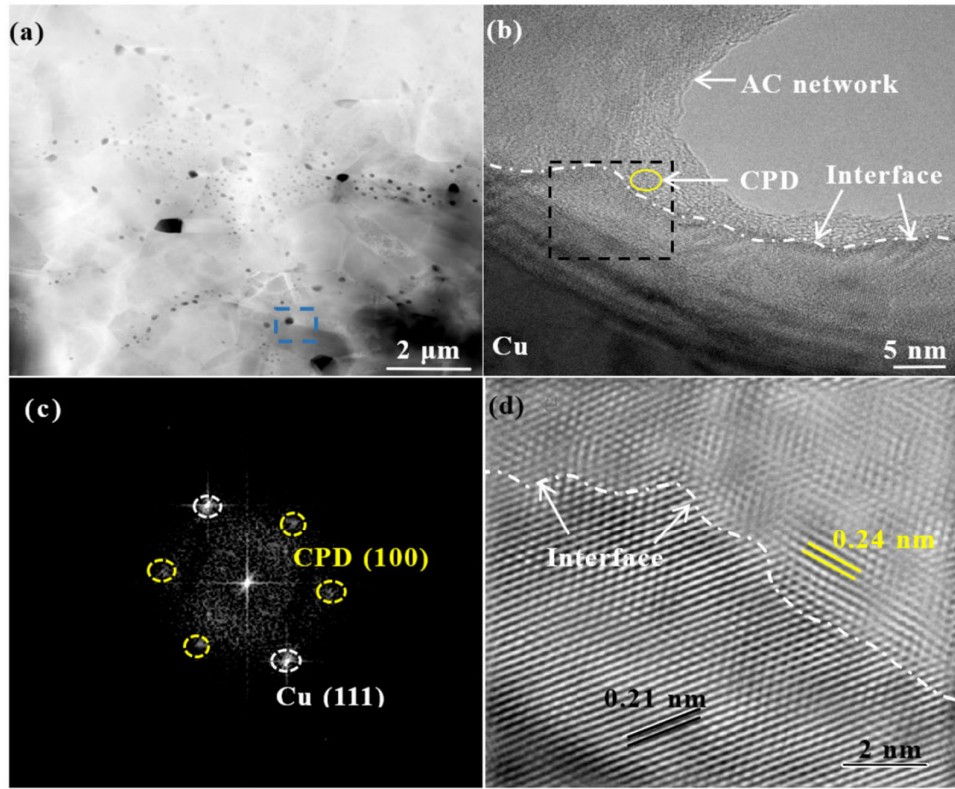

**Figure 3.** TEM image of *h3-CPD*/Cu composite (**a**), HRTEM image of *h3-CPD*/Cu (**b**), Fourier and inverse Fourier of the labeled wireframe in (**b**) of (**c,d**).

### 3.2. Mechanical Properties of Bulk Specimens

As shown in Table 1, the sintered bulk materials were near full densification. Figure 4a shows the stress–strain curves for the samples. Pure Cu had an ultra-tensile strength (UTS) of 260.2 ± 6.3 MPa and its elongation (EL) was 15.9 ± 0.7%. The *CPD*/Cu sample's UTS was 312.1 ± 7.0 MPa, and its EL was 34.3 ± 1.8%, which suggested the addition of *CPD* improved the mechanical property of the Cu matrix. For *h1-CPD*/Cu, the UTS was 268.1 ± 7.3 MPa, and the EL was 18.8 ± 3.5%. For *h2-CPD*, *h3-CPD*/Cu samples, the values for UTS and EL were enhanced, compared with pure Cu, and the values were 281.1 ± 0.8 MPa, 30.0 ± 4.4%; 283.0 ± 0.8 MPa, 34.5 ± 1.2%, which was mainly due to the semi-coherent interface formed between *h-CPD* and Cu. The semi-coherent interface promoted load transfer. In contrast to *CPD*/Cu composites, however, the surface functional groups and organic chains of *CPD* were decomposed to varying degrees during heat treatment. As a result, a significant amount AC was formed (as shown in Figure 3), lowering the mechanical properties of the composites. UTS (278.2 ± 4.2 MPa) and EL (16.1 ± 6.8%) of the *h4-CPD*/Cu sample were reduced to values near those of pure Cu.

**Table 1.** Relative density of Cu, *CPD*/Cu and *h-CPD*/Cu composites.

| Sample | Cu | *CPD*/Cu | *h1-CPD*/Cu | *h2-CPD*/Cu | *h3-CPD*/Cu | *h4-CPD*/Cu |
|---|---|---|---|---|---|---|
| Relative density | 98.8 ± 0.2 | 99.4 ± 0.1 | 99.5 ± 0.1 | 99.4 ± 0.1 | 99.0 ± 0.2 | 99.0 ± 0.3 |

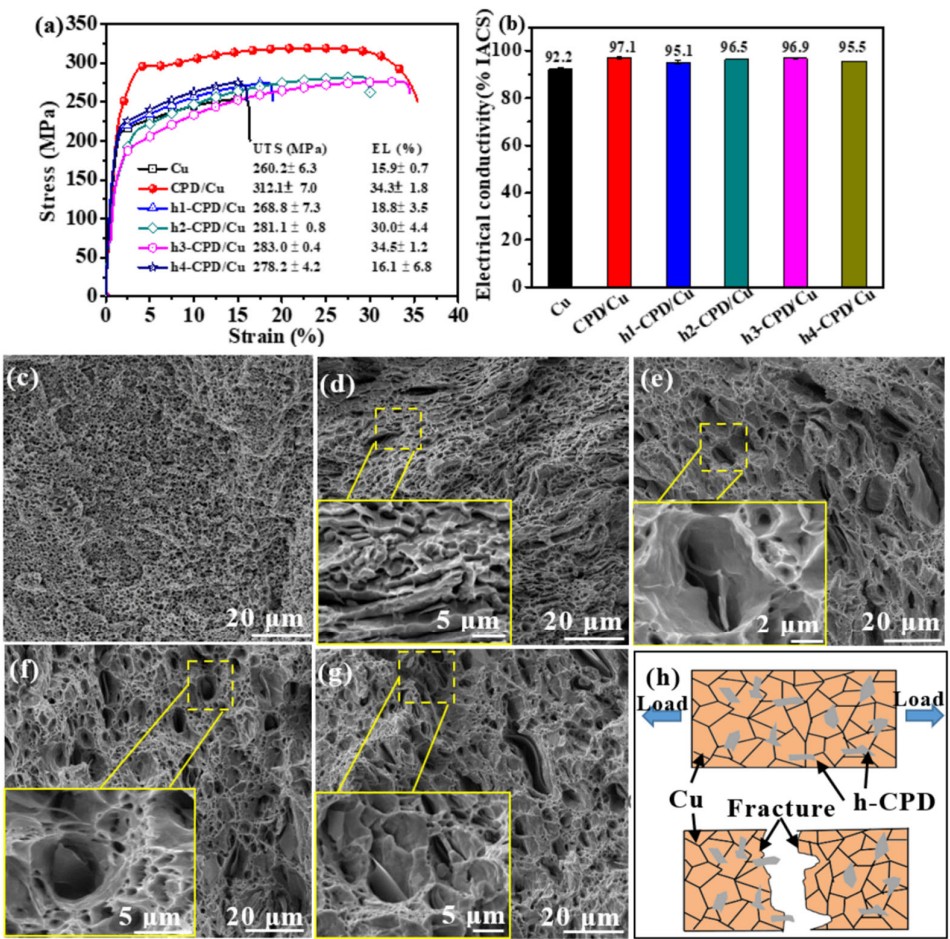

**Figure 4.** (**a**) stress–strain curves and (**b**) electrical conductivity of composites, fracture morphologies of (**c**) Cu, (**d**) *h1-CPD*/Cu, (**e**) *h2-CPD*, (**f**) *h3-CPD*, (**g**) *h4-CPD* composites, (**h**) fracture model of *h-CPD* reinforcement.

The electrical conductivities of the samples are shown in Figure 3b. Pure Cu was 92.2 ± 0.7%IACS. All Cu-based composites were higher than pure Cu. Specifically, *CPD*/Cu was 97.1 ± 0.4%IACS, *h1-CPD*/Cu was 95.1 ± 0.7%IACS, *h2-CPD*/Cu was 96.5 ± 0.2%IACS, *h3-CPD*/Cu was 96.9 ± 0.1%IACS, and *h4-CPD*/Cu was 95.5 ± 0.2% IACS. Both *CPD* and *h-CPD* had high conductivity due to the presence of pyridine-N, pyrrolic-N, and graphite-N. The reason for this was because following N doping, there was an increase in the density of unoccupied charge transport states, which showed that the charge transferred from C to the more electronegative N, improving the composites' conductivities [9,10]. Additionally, *CPD* functions well as an electron carrier itself [11]. However, once *CPD* was heated, the structure was disrupted, which resulted in a minor drop in the electrical conductivity of *h-CPD*/Cu composites.

Figure 4c–g shows the fracture morphology of samples after a tensile test. All samples showed ductile fracture properties, evidenced by the dimples on the fracture surface. Pull-out *h-CPD* could be seen on the fracture surface (Figure 4d,g), which suggested load transfer from the Cu matrix to reinforcements during tensile testing, in contrast to the fracture morphology of pure Cu (see Figure 4c). The *h-CPD* was clearly adherent and embedded in the Cu matrix, which was inseparable from the strong interface bonding.

## 4. Conclusions

In summary, *h-CPD*/Cu composites were prepared, together with pure Cu and *CPD*/Cu, under the same process. The semi-coherent interface increased *h-CPD*/Cu interface bonding, load transfer, and electron transport. However, *h-CPD* had a lot of AC, which formed via *CPD* surface functional groups and polymer chains decomposing during heat treatment, reducing the composite's mechanical and electrical properties. This study reveals that the structural integrity of *CPD* helps prepare composites with excellent comprehensive properties.

**Author Contributions:** Literature search, J.X.; manuscript writing, J.X.; study design, W.Z., R.B. and J.Y.; date analysis, W.Z. and S.L.; making charts, S.L.; date collection, R.B. and Z.L. All authors have read and agreed to the published version of the manuscript.

**Funding:** This research is funded by the National Natural Science Foundation of China Grant No. 52174345 and 52064032, Scientific Research Starting Foundation for Advanced Talents of Jiangxi University of Science and Technology Grant No. 205200100628.

**Institutional Review Board Statement:** Not applicable.

**Informed Consent Statement:** Not applicable.

**Data Availability Statement:** The date presented in this study are available on request from the corresponding author. The date is not publicly available due to privacy.

**Conflicts of Interest:** The authors declare no conflict of interest.

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
