# Peer review of "Effect of Structure of Carbonized Polymer Dot on the Mechanical and Electrical Properties of Copper Matrix Composites"

_metals, doi:10.3390/met12101701_

Round 1

Reviewer 1 Report

The authors investigated the effect of structure of carbonized polymer dot on the mechanical and electrical properties of copper matrix composites. The presented results are interesting, the manuscript could be accepted for publication in Metals. I have some comments and suggestions to improve the quality of this study as follows:

1. Relative density of the composites should be provided.

2. Compare the mechanical properties and electrical conductivity of the prepared composites with other Cu matrix composites.

3. Strengthening efficiency of carbonized polymer dots on the mechanical properties should be calculated and then compared with the other reinforcements like CNTs, Gr, SiC, TiC, etc

4. Enhancement in the tensile strength of the composites may be contributed by some strengthening mechanisms such as grain refinement, solid solution, dislocation density, and load transfer. The authors should use these strengthening mechanisms to calculate their contributions to the mechanical properties.

Author Response

Thank you for your comments. 

Reviewer 2 Report

The authors present in a short communication the results of the studies on heat treated CPD/Cu in comparison to CPD/Cu and Cu. The h-CPD were analyzed regarding their microstructures after different heat treatment temperatures. After 500 C most CPD structures are damaged.  The conclusions of the authors are based on TEM images, XPS measurements and studies of the mechanical properties. A brief description of the synthesis is also given.

All in all an understandable and comprehensible paper, which should be published.

Author Response

Thanks!

Reviewer 3 Report

The paper by Xu et al. describes the use of carbonized polymer dots (CPD) in the preparation of Cu matrix composite. The results show that after the addition of CPD to a Cu matrix, the strength and electrical conductivity of the obtained composite increased compared to those of base material. 

However, prior to publication, some adjustments should be made:

1) Moderate English corrections are required.

2) Some errors and misspellings should be corrected e.g., Carbon nanotube (line 26).

3) Lines 74-75: "single, distributed form but instead prefers to attach to one another" - do you mean aggregation/agglomeration?

4) Have you thought about performing TG-DTG-DSC for the obtained materials in the future? Such analysis combined with with the evolved gas analysis using Fourier transformed infrared spectroscopy (FTIR) and mass spectrometry would provide you more information about thermal behaviour of the obtained materials.

Author Response

Response to Reviewer 3 

The paper by Xu et al. describes the use of carbonized polymer dots (CPD) in the preparation of Cu matrix composite. The results show that after the addition of CPD to a Cu matrix, the strength and electrical conductivity of the obtained composite increased compared to those of base material. However, prior to publication, some adjustments should be made:

Point 1: Moderate English corrections are required.

Response 1: English corrections have been moderated. (in red)

Point 2: Some errors and misspellings should be corrected e.g., Carbon nanotube (line 26).

Response 2: The errors and misspellings have been corrected. (in red)

Point 3: Lines 74-75: “single, distributed form but instead prefers to attach to one another”-do you mean aggregation/agglomeration?

Response 3: Yes.

Point 4: Have you thought about performing TG-DTG-DSC for the obtained materials in the future? Such analysis combined with the evolved gas analysis using Fourier transformed infrared spectroscopy (FTIR) and mass spectrometry would provide you more information about thermal behavior of the obtained materials.

Response 4: Thank you for your good advice. TG-DTG-DSC test will be carried out on the obtained materials in the future.

Round 2

Reviewer 1 Report

The manuscript can be accepted for publication.